# Multiplex Proteomic Evaluation in Inborn Errors with Deregulated IgE Response

**DOI:** 10.3390/biomedicines11010202

**Published:** 2023-01-13

**Authors:** Enrico Scala, Stefania Madonna, Daniele Castiglia, Alessandro Scala, Elisabetta Caprini, Roberto Paganelli

**Affiliations:** 1Clinical and Laboratory Molecular Allergy Unit, IDI–IRCCS, 00167 Rome, Italy; 2Laboratory of Experimental Immunology, IDI–IRCCS, 00167 Rome, Italy; 3Laboratory of Molecular and Cell Biology, IDI–IRCCS, 00167 Rome, Italy; 4Fondazione Policlinico Universitario A. Gemelli IRCCS, Università Cattolica del Sacro Cuore, 00168 Rome, Italy; 5UniCamillus–Saint Camillus International University of Health Sciences, 00131 Rome, Italy

**Keywords:** inborn error, Comel–Netherton syndrome, recessive X–linked ichthyosis, Hyper–IgE, TSPAN7, MID1IP1, food allergy, atopic dermatitis, nsLipid transfer protein

## Abstract

(1) **Background:** Atopic dermatitis constitutes one of the most common inflammatory skin manifestations of the pediatric population. The onset of many inborn errors occurs early in life with an AD–like picture associated with a deregulated IgE response. The availability of proteomic tests for the simultaneous evaluation of hundreds of molecules allows for more precise diagnosis in these cases. (2) **Methods:** Comparative genomic hybridization microarray (Array–CGH) analysis and specific IgE evaluation by using allergenic microarray (ISAC) and microarray (ALEX2) systems were performed. (3) **Results:** Proteomic investigations that use multiplex methods have proven to be extremely useful to diagnose the sensitization profile in inborn errors with deregulated IgE synthesis. Four patients with rare diseases, such as recessive X–linked ichthyosis (RXLI, OMIM 308100), Comel–Netherton syndrome (NS, OMIM256500), monosomy 1p36 syndrome (OMIM: 607872), and a microduplication of Xp11.4 associated with extremely high levels of IgE: 7.710 kU/L, 5.300 kU/L, 1.826 kU/L, and 10.430 kU/L, respectively, were evaluated by micro– and macroarray multiplex methods. Polyreactivity to both environmental and food allergens was observed in all cases, including the first described case of association of X–chromosome microduplication and HIE. (4) **Conclusions:** Extensive use of proteomic diagnostics should be included among the procedures to be implemented in inborn errors with hyper–IgE.

## 1. Introduction

Atopic dermatitis (AD) constitutes one of the most common inflammatory skin manifestations, affecting up to 20% of the pediatric population and 10% of the adult population, associated with a complex interplay between genetic and environmental causal factors not all fully understood to date [1]. AD is a heterogeneous disease with various clinical manifestations (phenotypes) sustained by specific molecular endotypes. Several endotypes based on the age of onset, ethnic origin, and clinical features can be distinguished [2]. The endotype pattern of AD also includes extrinsic and intrinsic AD [2]. Extrinsic (allergic) AD represents approximately 80% of adult atopic patients and is associated with a high level of serum IgE. Elderly patients with this AD endotype show frequent allergic sensitization to airborne allergens and food allergens [3]. Intrinsic (nonallergic) AD is a less common subtype (≈20%); however, it affects the elderly in an increased proportion [4]. Intrinsic AD is characterized by normal or low serum IgE levels, the absence of atopic background, and a lack of sensitization to environmental allergens. However, specific IgE against enterotoxins of *S. aureus* and other microbial antigens have been identified [5].

Genetic pieces of evidence depict a complex network involving epidermal barrier dysfunction and dysregulation of innate and adaptive immunity in AD patients. GWAS and Immunochip analysis have identified a total of 19 susceptibility loci for AD. Mutations in the human filaggrin gene (FLG) are the most significant and well–replicated genetic mutations associated with AD. However, several studies demonstrated that FLG knockout alone is not sufficient to cause an AD–like phenotype in mice, suggesting the intriguing hypothesis that additional atopy–promoting gene variants are required to cause atopy in the context of FLG deficiency [6]. Indeed, other mutations associated with epidermal barriers such as SPINK5, FLG–2, SPRR3, and CLDN1 have all been linked to AD [7]. A recent whole–exome sequencing performed in a control group and a disease group with 20 AD patients without atopic march (AM) and 20 with AM suggests that polymorphism of DOCK8 and IL17RA might be related to the increase in the total IgE level [8].

On the other hand, some inborn errors often look similar to AD, both clinically and due to the finding of elevated levels of circulating IgE, which sometimes also leads to delays in proper diagnosis. This hyper–IgE (HIE) syndrome may also be associated with severe allergic phenomena, so it is essential to consider extensive in vitro allergy investigations in these cases, to unveil any hidden sensitizations that may, at various levels, impact the patient’s quality of life. HIE syndromes are a series of rare primary immunodeficiencies characterized by extremely high levels of IgE, AD–like eczema, and recurrent skin and/or lung infections (HIE triad), often associated with high mortality due to severe infections and high risk of malignancies. As mentioned, discriminating such forms from pictures of severe atopic dermatitis, especially in early childhood, can be very difficult. The most common form of HIE is caused by an autosomal dominant (AuD) negative mutation in the human signal transducer and activator of transcription 3 (*STAT3*) gene, which plays an essential role in signal transduction of several cytokines to their respective receptors (IL6, IL10, IL11, IL17, IL21 and IL22). STAT3–HIE, also known as Job’s syndrome, is associated with recurrent staphylococcal skin abscesses (cold abscesses), chronic mucocutaneous candidiasis (CMC), cystic pneumonia, and skeletal abnormalities [9].

An autosomal recessive (AR) form of HIE is caused by a deficiency of the dedicator of cytokinesis 8 (DOCK8). A DOCK8 deficit results in severely affected adaptive immunity and lymphocyte and dendritic migration; hence, the high susceptibility to viral skin and lung infections. T lymphocytes are Th2–imbalanced, with very high IgE levels and often severe food allergies, unlike what is observed in STAT3–HIE [10]. Comel–Netherton (CN) syndrome is caused by a mutation in the serine protease inhibitor of Kazal type 5 (*SPINK5*) gene that encodes for lymphoepithelial kazal type–related inhibitor (LEKTI), whose deficiency is associated with elevated desmoglein 1 degradation and subsequent stratum corneum detachment (causing congenital ichthyosis or, in other patients, a scaly erythroderma), trichorrhexis invaginata, also known as “bamboo hair,” and HIE [11]. The X–linked *FOXP3* gene codifies a transcription factor essential for the development of CD4+ T regulatory (Treg) cells. The absence of Treg cells prevents physiological regulation of the immune response, which will therefore be dysregulated on both the autoimmune and allergic response sides, thus causing the immunodysregulation polyendocrinopathy enteropathy X–linked (or IPEX) syndrome [12].

It is widely known that the prototypical STAT3–loss–of–function (LOF)–associated HIE syndrome is not associated with “allergic” phenomena despite the very high levels of circulating IgE due to both the reduced mast cell degranulation capacity in these patients and a likely lack of IgE affinity maturation due to a functional deficiency of T follicular helper 13 cells [10]. On the other hand, many of the abovementioned forms of HIE (DOCK8 deficit, Comel–Netherton, or IPEX syndrome) are associated with very severe forms of food, environmental, or latex allergies. The evaluation of the IgE specificity in these patients is therefore important for the management of these extremely rare conditions but also in more frequent conditions such as the typical AD syndrome. Nowadays, several proteomic methods are available in the market, such as specific IgE micro– and macroarray tests, which are extremely useful for evaluating hundreds of native or recombinant molecular components in patients with primary immunodeficiency (PID) [13,14] or AD [15].

Over the past 15 years, approximately 63,000 individuals referred to our institute, an Italian reference centre for dermatology and allergology and member of the European Reference Network on Rare and Undiagnosed Skin Disorders (ERN–skin), have been evaluated by proteomic methods for specific IgE assay. Of these individuals, a few dozen were affected by rare forms of inborn errors (often with a prevalence <1/1,000,000 inhabitants) characterized by deregulated IgE synthesis, in the context of immunodeficiencies of the most disparate kinds. Previous studies have documented the sensitization proteomic profile in these patients [13,14], but others have been recently identified and typed, thus providing further excellent examples to show how a multiplex approach in allergology proteomics has become an essential tool to investigate, evaluate, treat, and help these extremely rare conditions.

## 2. Materials and Methods

### 2.1. Recruitment, Characterization of the Patients, and Study Design

The study participants (Table 1) were enrolled between 2005 and 2022 at the outpatient Unit of IDI–IRCCS in Rome, a National Reference Center for Allergic and Dermatological diseases. Demographic details together with clinical data (food–related reactions, respiratory and dermatological symptoms) were recorded using the TD–Synergy^^®^^ Laboratory Information System (Siemens Healthcare Diagnostics, Muenchen, Germany) and a customized electronic database. Sera from the participant were collected and stored at the time of the visit.

### 2.2. Genomic Analysis

#### 2.2.1. Genomic Hybridization Microarray (Array–CGH) Analysis

Based on the presence of multiple congenital anomalies and dysmorphisms, a comparative genomic hybridization microarray (Array–CGH) analysis using the CytoChip oligo–array ISCA 4X44K resolution (BlueGnome, Cambridge, UK) was performed according to the recommendations of the manufacturer. Data were then analyzed by using the Blue–Fuse for microarrays software package (BlueGnome).

#### 2.2.2. SPINK5 Molecular Analysis and LEKTI Expression

Mutation screening was performed on blood genomic DNA by amplification and bidirectional Sanger sequencing of SPINK5 exons and flanking intronic borders (GenBank NM_006846.3), as described [16]. LEKTI expression was evaluated by immunohistochemistry on skin sections from formalin–fixed paraffin–embedded patients’ skin biopsies by using anti–LEKTI polyclonal antibodies directed to the D7D12 or D13D15 C–terminal regions, as described [17].

### 2.3. IgE Antibody Measurements

The blood samples were stored during the control visits from 2012 to 2022 for the evaluation of IgE reactivity by using ImmunoCAP ISAC^®^ (Thermo Fisher Scientific, Sweden), and Allergy Explorer–ALEX2^®^ (Macroarray Diagnostics, Vienna, Austria).

#### 2.3.1. ImmunoCAP ISAC^®^

In brief, ISAC slides were washed, rinsed, and dried at room temperature (RT). Undiluted serum (30 μL) from each patient was pipetted onto the slide and after 120 min incubation at RT in a humid chamber, slides were washed, rinsed, and dried. IgE binding was detected by the addition of an antihuman secondary antibody. Slides were then washed, rinsed, dried, and stored in the dark until scanning. Images were acquired by scanning allergen biochips with a CapitalBioLuxScan™ 10K microarray scanner. IgE values were expressed as ISU arbitrary units (ISAC Standardized Units) corresponding to IgE antibody levels in the ng/mL range (detection limit: 0.3 ISU–E).

#### 2.3.2. Allergy Explorer–ALEX2^®^ Macroarray Analysis

In the ALEX test, 300 allergens, including molecules and extracts, are spotted onto a nitrocellulose membrane in a cartridge chip, which is then incubated with 0.5 mL of a 1:5 dilution of serum, containing a CCD inhibitor under agitation. After incubation for two hours, the chips are washed three times and a pretitred dilution of antihuman IgE labelled with alkaline phosphatase is added and incubated for 30 min. After another cycle of extensive washing, the enzyme substrate is added, and after eight minutes, the reaction is stopped. The membranes are dried, and a charge–coupled device camera measures the intensity of the colour reaction for each allergen spot. The dedicated software digitalizes the images and prepares a report that lists the allergens and components and their score in kUA/mL. Finally, an arbitrary calibration curve is obtained by reacting four spots with decreasing concentrations of specific IgE corresponding to <0.35, 0.35–1, 1–5, 5–50 kUA/L and >50 kUA/L. We considered positive a concentration of ≥0.3 kUA/L.

## 3. Results

### 3.1. Recessive X–Linked Ichthyosis (RXLI, OMIM: 308100)

X–linked recessive ichthyosis (RXLI), the second–most common form of ichthyosis (estimated prevalence of 1–5/10,000), is a benign form of ichthyosis that affects almost exclusively males and is inherited through carrier females. RXLI is an epidermal lipid metabolism defect due to inactivating mutations or deletions in the steroid sulfatase STS gene (Xp22.3), thus resulting in disorders of cornification, mainly localized on extensor and flexor areas of the extremities, chest, and neck (“dirty neck” appearance) [18].

A 12–year–old male patient with RXLI already identified in another centre came to our observation as suffering from a severe form of atopic dermatitis accompanied by elevated levels of IgE (7710 kU/L). After evaluation with macroarray proteomics (Figure 1), the following reactivities were shown: food allergy driven by nsLipid transfer protein sensitization, and reactivity towards pollen allergens such as cypress, olive tree, and grass pollens. Interestingly, an allergy to yellow jackets (Vespula vulgaris) was also recorded. Figure 1.

Interestingly, very high levels of IgE directed against cross–reactive carbohydrate determinants (CCD) were observed in this case (Hom s Lactoferrin | CCD marker [19]), underscoring the importance of multiplex evaluation with molecular components in these complex cases. CCDs are ubiquitous oligosaccharide side chains present in various pollen and food glycosylated molecules and most of the glycoproteins found in Hymenoptera venoms. Approximately 25% of patients with bee venom allergy and approximately 10% with Vespula venom allergy have positive anti–CCD IgE values. Furthermore, more than 30% of patients with AD scored positive for CCD [15]. It is important to emphasize that anti–CCD IgE is responsible for test positivity, but is not able to induce any clinical symptoms [19]. The latest version of the ALEX^2^ test, currently available in the market can eliminate 60–70% of cases or at least reduce the CCD signal, using a specific inhibitor.

### 3.2. Comel–Netherton Syndrome (NS, OMIM: 256500)

Comel–Netherton syndrome is characterized by an ichthyosiform AD–like pattern, a bamboo–like hair–shaft defect and multiple atypical manifestations, caused by SPINK5 mutations [20].

A 52–year–old patient with NS, carrying the SPINK5 mutation c.238insG (ex4)/nonsense R217X (ex 8), associated with absent expression of LEKTI, came to our observation presenting a scaly erythroderma cutaneous form and complaining of violent postprandial abdominal pains. Elevated IgE levels (5300 kU/L) were observed.

Proteomic diagnostics (Figure 2) demonstrated the presence of reactivity to environmental allergens such as dust mites, pellitory of the wall, and male cats.

The patient was also allergic to several food allergens such as nsLTP (peach, maize, and wheat), arginine kinase (thermolabile molecule of crustaceans), and parvalbumins from fish (molecules resistant to cooking and gastric digestion, thus causing severe allergic reactions) (see Figure 2).

A diet eliminating the foods that emerged from the microarray test had determined an immediate and marked improvement in the patient’s cutaneous picture, and gastrointestinal symptoms, with a great improvement in the perceived quality of life.

### 3.3. Monosomy 1p36 Syndrome (OMIM: 607872)

The 1p36 deletion syndrome is characterized by multiple congenital abnormalities (i.e., heart disease, short stature, brachi– or microcephaly, ear dysplasia, hearing impairment or deafness, facial clefts) and mental retardation [21]. Deletion 1p36 is the most common terminal deletion syndrome in humans, with an estimated prevalence of approximately 1/5000 [22].

A 20–year–old patient affected by the 1p36 deletion syndrome caused by a cr.1 short–arm microdeletion in the p36.33p36.32 region, already identified at birth in another centre, was affected by a cognitive and stature developmental delay, ear asymmetries, sunken eyes, flat nasal bridge, and chronic inflammatory polyarthritis. Cognitive retardation was associated with communication and relationship difficulties, motor stereotypies, bilateral convergent strabismus, locomotion difficulties, ataxic gait, cingulate hypotonia, and easy fatigability. The girl communicated with gestures, complained continuously, especially at night, and often showed urticarial rash accompanied by severe itching, so she was placed on continuous antihistamine therapy.

Routine blood tests showed elevated IgE levels (1826 kU/L), and therefore the patient was accompanied to our out clinics, where a proteomic test was performed (Figure 3), showing polyreactivity to a variety of environmental (Alternaria alternata, dog epithelium, cypress tree, and grasses pollen), and food allergens (nsLipid transfer protein) (see Figure 3).

The modification of the dietary regimen operated based on the molecular tests, with the exclusion of wheat and fruit, resulted in the total disappearance of the symptoms that had been afflicting the young patient for several years, showing the usefulness of a multiplex approach even in those genetic defects characterized by a neurodevelopmental disorder with or without anomalies of the brain, with poor or absent speech, and inability to collect a correct clinical history.

### 3.4. Microduplication of Xp11.4 in a Girl with a Cognitive Defect, Cerebellar Hypoplasia, Atopic Dermatitis, and Hyper–IgE Syndrome

A 7–year–old woman presented a clinical picture characterized by cognitive defect, overweight, cerebellar malformation, (cerebellar hypoplasia left hemisphere), bilateral clubfoot (undergone surgery), AD, marked keratosis pilaris, acanthosis nigricans, thick lips, hair anomalies (frizzy, breaking ends), HIE syndrome, was the firstborn of a healthy, young and unrelated couple, and had a healthy sister two years younger. No family history of genetic diseases was reported. The patient was delivered at 39 weeks of gestation by caesarean section. Her birth weight was 3 kg and her length was 52 cm. She was breastfed for three months.

The genomic analysis identified a microduplication of 535 kb on chromosome Xp11.4, bridging from 38,561,541 bp to 39,096,541 bp, according to the human genome NCBI Build 38 version [10]. Karyotype analysis resulted in a normal 46, XX. The Xp11.4 duplication involved two known genes: (i) TSPAN7, and (ii) MID1 interacting protein 1 (MID1IP1) (Table 2). The microduplication was probably inherited from the father, because the younger sister, apparently healthy, was also a carrier of this microduplication.

The patient had severe AD, associated with hyper–IgE (IgE > 10,000 kU/L) and sensitization to various environmental allergens. Serial evaluations carried out with multiplex methods (both ImmunoCAP ISAC^®^ microarray containing an increasing number of allergens tested from 79 to 103, 112, and, more recently, 295 with the ALEX2^^®^^ macroarray), showed an IgE polysensitization picture, including environmental, both indoor (house dust mites and moulds) and outdoor (grasses, pellitory, and Oleaceae), and food allergen reactivity (Figure 4). The extensive reactivity to even minor dust mite molecules, such as Der p 7 and Der p 23, was not associated with severe respiratory symptoms or asthma. Similarly, LTP reactivity was never associated with a severe immediate adverse reaction. Interestingly, IgE reactivity to Ole e 9 and Asp f 6 was found, components already considered markers of atopic dermatitis in the Mediterranean area [15].

The serial IgE evaluations carried out during the follow–up visits showed a substantial constancy of the sensitizations recorded, except for some molecules gradually becoming available in the most recent tests, including Der p 23, a faecal molecule of mites, which is particularly associated with the possible onset of bronchial asthma (Figure 4). In our case, the patient had never suffered from severe respiratory symptoms, in line with what we have already observed in AD, where sensitization to molecules usually responsible for severe respiratory and food allergy pictures in allergic people, is associated with less severe clinical pictures [15].

Similarly, in the case of hyperreactivity to nsLTPs, this sensitization did not cause immediate symptoms, but the worsening of the skin condition in the days following the intake of culprit foods.

## 4. Discussion

The assessment of specific IgE in inborn errors with HIE allows the establishment of an actual IgE sensitization profile. The use of micro– or macroarray is helpful either to exclude other forms of HIE disorders, such as STAT3–LOF, where increased IgE synthesis is not followed by specific recognition of environmental or food allergens [10] or to accomplish the serial monitoring of these diseases. This is particularly important even in pediatric cases where the availability of biological material for such assessments is often scarce, so methods capable of evaluating hundreds of components using 50 to 100 microliters of serum are extremely useful [14].

In patients studied, we observed a pattern of polysensitization frequently found in AD. Interestingly, all patients studied were reactive to lipid transfer proteins, molecules frequently associated with an allergy to Rosaceae fruits, nuts, hazelnuts, and legumes in the Mediterranean area [23]. On the other hand, peculiar pictures of reactivity were recorded in the patients herein reported. Similarly, in the case described with 1p36 deletion syndrome, the abolition of LTP–related foods (fruits and wheat) from the diet resulted in a marked improvement in the patient’s clinical condition and quality of life.

In one case (RXLI), reactivity to cross–reactive carbohydrate determinants was evidenced. This sensitization, devoid of any clinical impact, may result in false in vitro reactivities capable of, if not identified, fouling the diagnostic data and inducing erroneous behavioural interventions [19]. This finding confirms, once again, the importance of diagnostic systems capable of highlighting all possible sensitization profiles, relevant and irrelevant.

In another case (microduplication of Xp11.4), reactivity to autoreactive (SOD) allergenic components was shown. Manganese superoxide dismutase (MnSOD), which may play a role as an autoantigen (molecular mimicry between fungal and human MnSOD) in the context of an autoimmune reactivity, can elicit a specific IgE response in a subset of patients with atopic dermatitis [24]. This patient, to the best of our knowledge, is the first reported case of HIE associated with an X–chromosome microduplication. Interestingly, our patient was reactive to both Asp f 6 and Ole e 9 considered to be the signature of patients with atopic dermatitis in the Mediterranean area [15].

The patient suffering from monosomy 1p36 Syndrome had no inborn error correlated with altered IgE response [25], so she had never been studied for allergy–mediated diseases. However, in this case, allergic sensitization to several nsLTP–related food allergens severely impacted the quality of life of the young patient, who was unable to correctly express the real reason for her problems. In this case, the use of a proteomic approach succeeded in clarifying the patient’s reactivity profile, confirming the usefulness of this approach in comprehensively defining the allergic status of patients with cognitive impairment.

In this study, in addition to inborn errors already known, such as NS, 1p36 deletion syndrome, and RXLI, we reported a case of micro–duplication of Xp11.4 involving 2 genes, TSPAN7 and MID1IP1. TSPAN7 (also known as CD231, A15, TALLA–1) is a member of the tetraspanin superfamily, cell–surface proteins found in all multicellular eukaryotes. Genetic defects of the relative coding gene (TM4SF2) are associated with X–linked intellectual disability [26]. However, tetraspanins are expressed in a wide variety of cell types and have functional roles in the regulation of cell development, activation, growth and motility and may have a role in the control of neurite outgrowth [27]. In immune cells, it is estimated that at least 20 different members of the tetraspanin superfamily are expressed on the leucocyte surface where they organize specific receptor and signalling proteins into functional microdomains in the plasma membrane [28]. Tetraspanins interact with key leucocyte receptors, including MHC molecules, integrins, CD4/CD8 and the B–cell receptor complex and, as such, can modulate leucocyte receptor activation and downstream signalling pathways [29,30]. The expression of the tetraspanins such as CD9, CD37, CD53, CD63, CD81, and CD82 on FcεRI^pos^ skin dendritic cells (DCs) from atopic dermatitis patients is significantly higher when compared to skin DCs of not allergic healthy individuals [31]. Furthermore, CD81 on B cells induces IL–4 and antibody production during Th–2 immune responses [32]. On the other hand, the function of MID1IP1 has not yet been fully elucidated, but MID1 plays important role in TH2 cell development via STAT6–dependent pathways [33]. Lyon’s hypothesis suggests that in mammals, all excess X chromosomes are functionally inactivated from the earliest stages of embryogenesis. Because of this, females having any partial Xp duplications are usually carriers of a normal phenotype [34,35]. In the literature, there are very few reports of interstitials of the short arm of the X chromosome, which are not accompanied by other chromosomal abnormalities [36]. As already reported, due to the well–known “phenomenon of inactivation of the X chromosome” women with partial Xp duplications, in many cases, have a completely normal phenotype, whereas only some may show a variable phenotype [34]. Even in our case, the sister of our patient, despite having a similar microduplication of Xp11.4, was clinically healthy, nonatopic, and phenotypically normal.

When Xp duplication occurs, doubled genes can cause functional disomy of usually inactivated genes in normal females with XX karyotype. In the patient we describe, two genes were present in the duplicated 535 kb microsegment. One of them has already been deemed responsible for X–linked mental retardation (TSPAN7), but little is known about MID1IP1 in the literature, except for an association with rye allergy [37]. Moreover, the incorrect allocation of these genes, the positional effects or the altered regulation of the sequences at the breakpoints of microduplication might have affected the temporal and spatial expression of these genes influencing also the development and functioning of the immune system. We, therefore, may assume that the duplication of these genes could be associated with cognitive defects and allergic phenotype with severe AD and HIE in our patient. The healthy phenotype of the sister, despite the presence of the same microduplication, points to a possible alteration of the mechanism of X chromosome inactivation.

Possible limitations to the use of proteomic methods for the multiplex evaluation of IgE reactivity are the high costs and the need for specific expertise for the correct interpretation of results, which must always be correlated with the clinical condition of the individual patient. Future aspects are represented by the continuous evolution of these approaches, which have seen a tripling of the components available for diagnosis in the space of a few years, making it possible to increasingly refine the diagnostic precision and impact in daily clinical practice, even in such rare diseases.

## 5. Conclusions

In conclusion, the new methods currently available to study IgE sensitization in vitro through the use of molecular components organized in micro– or macroarrays represent an important innovation in the diagnosis of even the rarest diseases, such as inborn errors with HIE. They enable us to not only verify whether circulating IgE have matured their affinity and are responsible for allergic sensitization, but also, if this is the case, to establish precisely the sensitizations present, even in cases in which the patient’s cognitive defect does not allow for correct allergological classification. This approach improves both the management of skin pathology and the quality of life of individual patients. Finally, investigations on genetic variants of risk for AD in patients with inborn errors associated with HIE could help to understand their susceptibility to manifest AD–like clinical symptoms [38].

## Figures and Tables

**Figure 1 biomedicines-11-00202-f001:**
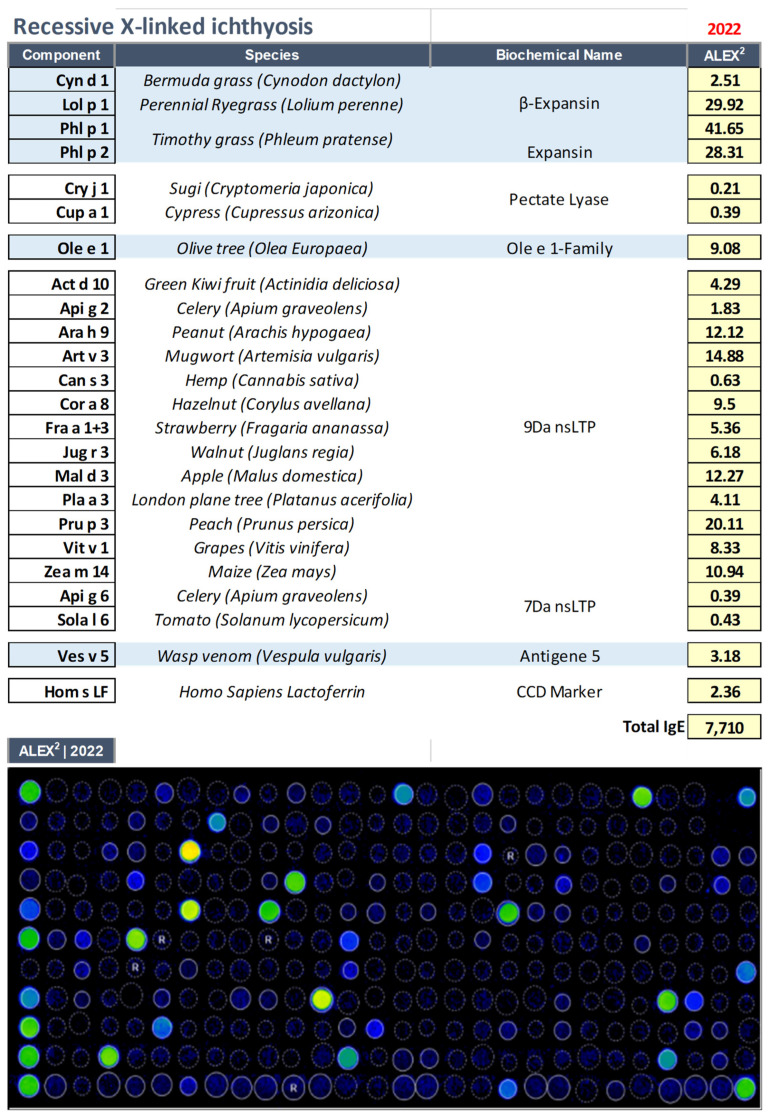
IgE sensitization profile as evaluated through multiplexed proteomic evaluations with Allergy Explorer–ALEX2^®^ 2 macroarray in the patient affected by recessive X–linked ichthyosis (RXLI, OMIM: 308100).

**Figure 2 biomedicines-11-00202-f002:**
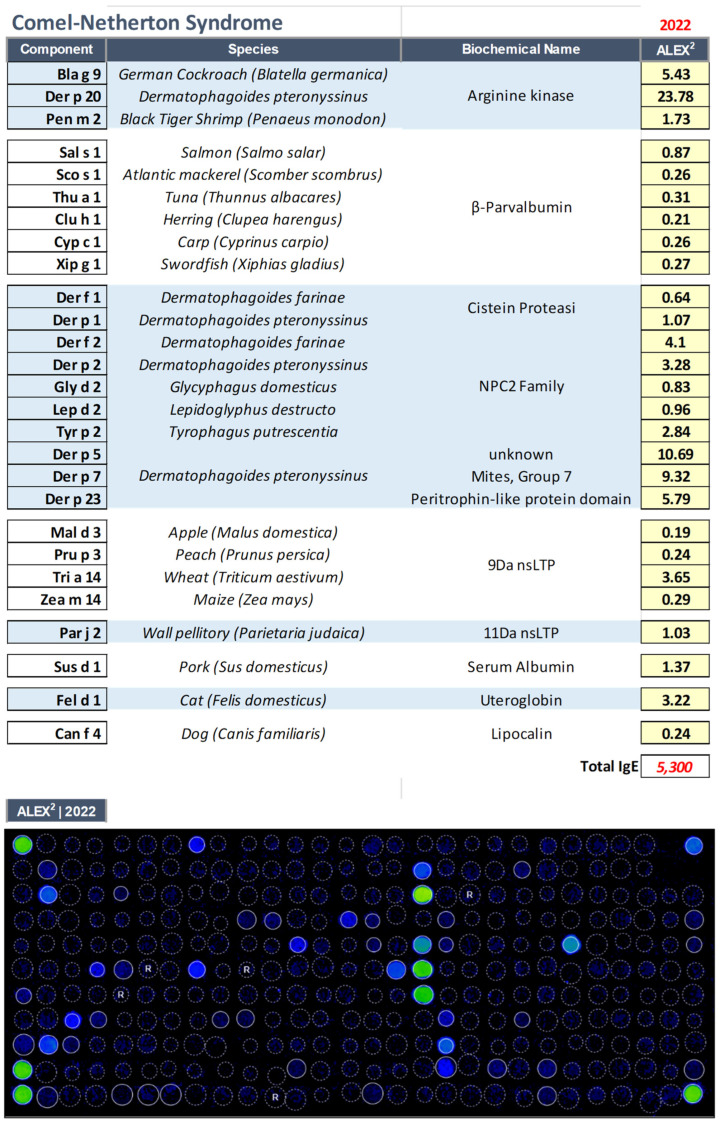
IgE sensitization profile as evaluated through multiplexed proteomic evaluations with Allergy Explorer–ALEX2^®^ 2 macroarray in the patient affected by Comel–Netherton Syndrome (NS, OMIM: 256500).

**Figure 3 biomedicines-11-00202-f003:**
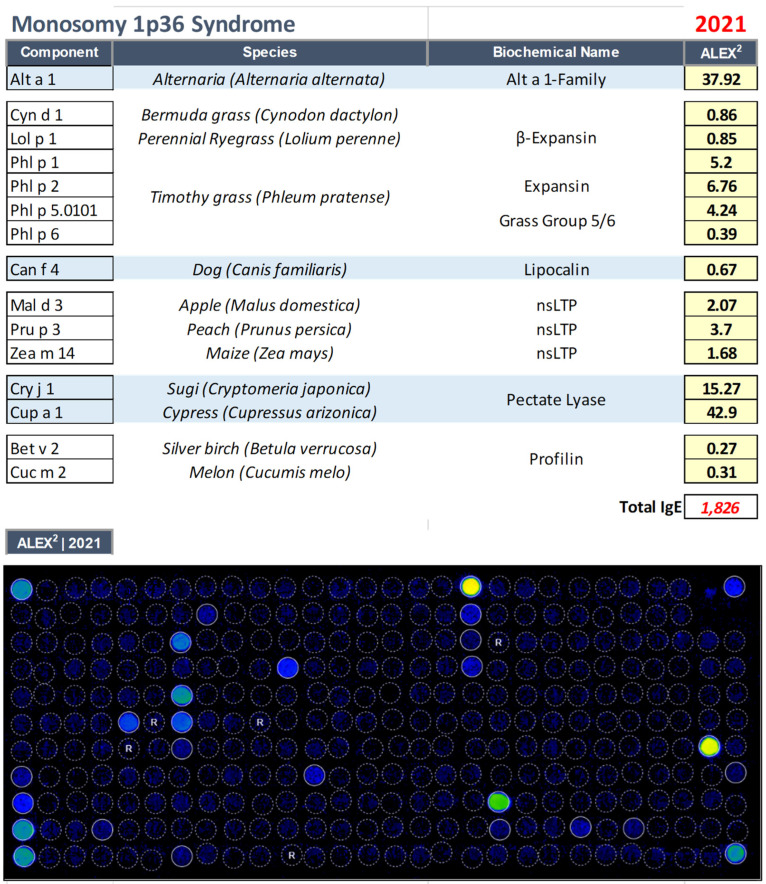
IgE sensitization profile as evaluated through multiplexed proteomic evaluations with Allergy Explorer–ALEX2^®^ 2 macroarray in the patient affected by Monosomy 1p36 Syndrome (OMIM: 607872).

**Figure 4 biomedicines-11-00202-f004:**
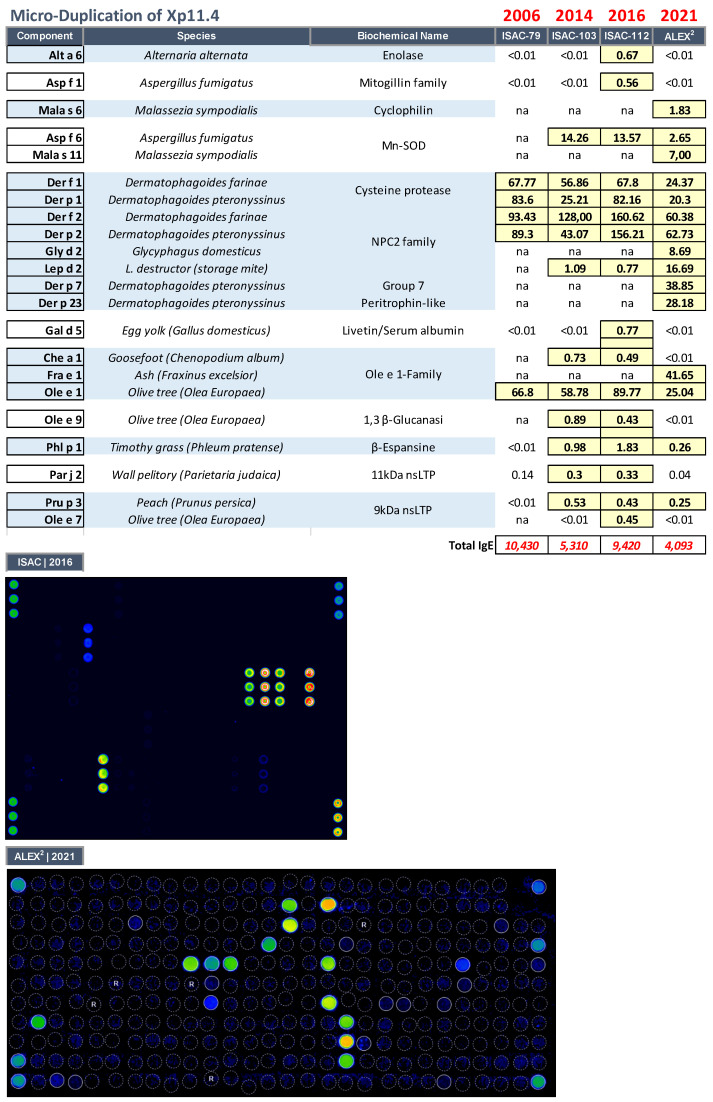
IgE sensitization profile as evaluated through multiplexed proteomic serial evaluations with ImmunoCAP ISAC^®^ microarray (2006, 2014, and 2016) and Allergy Explorer–ALEX2^®^ macroarray (2021), in the patient affected by a microduplication of Xp11.4.

**Table 1 biomedicines-11-00202-t001:** Patients with inborn error and impaired IgE production under study.

Age	Sex	Disease
12 yrs	Male	*Recessive X–linked ichthyosis* (RXLI, *OMIM: 308100*)
52 yrs	Male	*Comel–Netherton Syndrome (NS, OMIM: 256500)*
20 yrs	Female	*Monosomy 1p36 Syndrome (OMIM: 607872)*
7 yrs	Female	*Microduplication of Xp11.*

**Table 2 biomedicines-11-00202-t002:** Online Mendelian Inheritance in Man (OMIM^®^)–Gene–Map–List on chromosome–X.

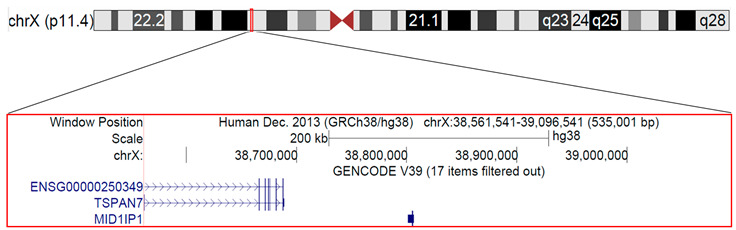								
**Cytogenetic location**	**Genomic coordinates (NCBI/GRCh38)**	**Gene Locus**	**Gene Locus name**	**Gene Locus MIM number**	**Approved Symbol**	**Entrez Gene ID**	**Phenotype**	**Inheritance**
Xp11.4	X:38561541-38688917	TSPAN7, TM4SF2, MXS1, A15, XLID58	Tetraspanin 7	300096	** TSPAN7 **	7102	Intellectual developmental disorder, X-linked 58	X-linked recessive
Xp11.4	X:38801458-38806531	MID1IP1, MIG12	MID1-interacting protein 1	300961	** MID1IP1 **	58526		

## Data Availability

The data that support the findings of this study are available on request from the corresponding author. The data are not publicly available due to privacy or ethical restrictions.

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
