# Peer review of "Multiplex Proteomic Evaluation in Inborn Errors with Deregulated IgE Response"

_biomedicines, 2023, doi:10.3390/biomedicines11010202_

Round 1

Reviewer 1 Report

This study demonstrates the use of proteomic diagnostics to study IgE sensitization in patients with rare diseases, such as inborn errors with hyper-IgE. In two of the four cases presented, the results from the macroarray were used to modify dietary regimens and improve quality of life. 

This is an interesting study, however, requires some improvements before publication. 

Specific Comments: 

- Figure 1 is of poor quality (blurry) and thus cannot be seen or interpreted. Please improve quality and size so that the figure is readable. 

- It would be good to have a figure on the analysis of the IgE values for each patient (the ImmunoCAP ISAC is included in the methods but there is no figure) 

- The introduction reads more like a review; there is a lot of information that is irrelevant to the results of this paper, such as the explanation of a lot of diseases that the patients included in this study do not have. Please be more concise and include only information relevant to the current study. 

- Some sentences need to be rewritten for clarity and improved grammar. 

Line 70: Please rewrite this sentence, it is unclear what is meant by "AD-like picture" - symptoms? genetic polymorphisms? 

It may also be clearer for each patient to have a table or list of their reactivities rather than all in a giant paragraph as it is difficult to follow. 

Line 223: Please rewrite for clarity - how do you compensate for the false reactivity? It is unclear what this sentence means. 

Author Response

Point-by-point response template
Date: 23 dec 2022
Manuscript Number: biomedicines-2108364
Title of Article: Multiplex proteomic evaluation in inborn errors with deregulated IgE response
Name of the Corresponding Author: Enrico Scala
Email Address of the Corresponding Author: [email protected]

Dear Editor

Thank you for the relevant and thorough review of our manuscript, and for the very kind invitation to submit a revised manuscript for consideration as a full article. We have addressed all the comments that were raised by the reviewers, and this has improved the manuscript 
considerably. Therefore, we now submit a revised manuscript that we hope is suitable for publication in biomedicines

Specific Responses:

Dear reviewer
Thank you for your valuable time and comments to the manuscript. We appreciate your efforts

reviewer

This study demonstrates the use of proteomic diagnostics to study IgE sensitization in patients with rare diseases, such as inborn errors with hyper-IgE. In two of the four cases presented, the results from the macroarray were used to modify dietary regimens and improve quality of life.

This is an interesting study, however, requires some improvements before publication.

Specific Comments:

- Figure 1 is of poor quality (blurry) and thus cannot be seen or interpreted. Please improve quality and size so that the figure is readable.

- It would be good to have a figure on the analysis of the IgE values for each patient (the ImmunoCAP ISAC is included in the methods but there is no figure)

*****Thank you for this relevant comment.
We have totally changed Figure 1, breaking it down into 4 parts (one for each patient examined), inserting for each patient the molecules tested positive, thus following the suggestions of this, and similar ones of other reviewers.
We have added, as requested, an image of the ISAC test (Figure 1d)

reviewer

- The introduction reads more like a review; there is a lot of information that is irrelevant to the results of this paper, such as the explanation of a lot of diseases that the patients included in this study do not have. Please be more concise and include only information relevant to the current study.

*****Thank you for this pertinent observation. 
We have profoundly reduced the introduction, eliminating irrelevant aspects and reducing the description of the different immunodeficiencies associated or not associated with the presence of allergies, despite the high IgE levels recorded and the atopic dermatitis-like appearance.

reviewer

- Some sentences need to be rewritten for clarity and improved grammar.

Line 70: Please rewrite this sentence, it is unclear what is meant by "AD-like picture" - symptoms? genetic polymorphisms?

*****We have rewritten the sentence as suggested by the reviewer (now lines 65-67)

reviewer

It may also be clearer for each patient to have a table or list of their reactivities rather than all in a giant paragraph as it is difficult to follow.

******We followed the reviewer's suggestion including the specifications of the recorded reactivities directly into a table associated with each patient's figure (Figure 1 a - d)

reviewer

Line 223: Please rewrite for clarity - how do you compensate for the false reactivity? It is unclear what this sentence means.

******We thank the reviewer for pointing this out. We have supplemented the information provided by adding a further explanation on the issue of CCD and false-positive signals induced by these glycosylated allergens. (lines 191 - 199)

Reviewer 2 Report

Very interesting work on the study of diseases with elevated IgE that simulate atopic dermatitis (AD). They are certainly included in the differential diagnosis of AD. Proteomics opens a window of opportunity to refine the study of these patients, which would ideally be available in all centres. The article is well done, I have no comments for the authors. 

Congratulations

Author Response

Dear reviewer
Thank you for your valuable time and comments on the manuscript. We appreciate your efforts

reviewer

*****We thank the reviewer for his/ her supportive comments.

Reviewer 3 Report

In this study, Scala and colleagues investigated the proteome of inborn errors with dysfunctional IgE response in AD. Authors used micro and macro protein array to the study the proteome sensitization profile of 4 patients with rare mutations- recessive X-linked ichthyosis (RXLI, OMIM 308100), Comel Netherton syndrome (NS, OMIM256500), monosomy 1p36 syndrome (OMIM: 607872), and a micro-duplication of Xp11.4. This study is relevant in the AD field and provides great platform to study rare mutation in AD.

·       Wherever it is possible, provide pictures of clinical and histological manifestation of before and after treatment of the patients studied here.

·       Describe future aspect and limitation of this study in discussion.

·       The manuscript is written in a review style particularly introduction section; change this to suit the article type (research article).

·       Include a table describing the age, gender and disease type for all patients.

·       Informed consent section states that consent was obtained from three patients. What about the fourth patient?

·    Resolution of figure 1 is not great. Replace this figure with better resolution image. 

Author Response

Dear reviewers.
Thank you for your valuable time and comments on the manuscript. We appreciate your efforts

reviewer
Comments and Suggestions for Authors
In this study, Scala and colleagues investigated the proteome of inborn errors with dysfunctional IgE response in AD. Authors used micro and macro protein array to the study the proteome sensitization profile of 4 patients with rare mutations- recessive X-linked ichthyosis (RXLI, OMIM 308100), Comel Netherton syndrome (NS, OMIM256500), monosomy 1p36 syndrome (OMIM: 607872), and a micro-duplication of Xp11.4. This study is relevant in the AD field and provides great platform to study rare mutation in AD.

·       Wherever it is possible, provide pictures of clinical and histological manifestation of before and after treatment of the patients studied here.

******We thank the reviewer for the suggestion, but unfortunately we do not have such material available for publication

reviewer

·       Describe future aspect and limitation of this study in discussion .

******We have added a specific paragraph at the end of the discussion to highlight these interesting aspects, as suggested by the reviewer (lines 385 - 391)

reviewer

·       The manuscript is written in a review style particularly introduction section; change this to suit the article type (research article).

******We have reduced and made the introduction more concise, focusing on the issue of atopic dermatitis and those inborn errors associated with high levels of IgE with or without associated clinically impacting allergological reactivity

reviewer

·       Include a table describing the age, gender and disease type for all patients.

******Added a new Table (line 171)

reviewer

·       Informed consent section states that consent was obtained from three patients. What about the fourth patient?

******Sorry for the mistake, We have fixed this point

·    Resolution of figure 1 is not great. Replace this figure with better resolution image

******We have totally changed Figure 1, as requested also by another reviewer, inserting for each patient the molecules found positive

Round 2

Reviewer 1 Report

Thank you for responding to my comments; the manuscript is much clearer and highlights the nice work done by the authors. I am happy for this to be published. 

Reviewer 3 Report

Responses provided by the authors are adequate.